# Cancer Stem Cells and Androgen Receptor Signaling: Partners in Disease Progression

**DOI:** 10.3390/ijms242015085

**Published:** 2023-10-11

**Authors:** Juan Carlos Quintero, Néstor Fabián Díaz, Mauricio Rodríguez-Dorantes, Ignacio Camacho-Arroyo

**Affiliations:** 1Unidad de Investigación en Reproducción Humana, Instituto Nacional de Perinatología-Facultad de Química, Universidad Nacional Autónoma de México, Mexico City 11000, Mexico; jc.chimiste@gmail.com; 2Departamento de Fisiología y Desarrollo Celular, Instituto Nacional de Perinatología, Mexico City 11000, Mexico; nfdiaz00@yahoo.com.mx; 3Laboratorio de Oncogenómica, Instituto Nacional de Medicina Genómica, Mexico City 14610, Mexico

**Keywords:** cancer stem cell, androgen receptor, prostate cancer, breast cancer, glioblastoma

## Abstract

Cancer stem cells exhibit self-renewal, tumorigenesis, and a high differentiation potential. These cells have been detected in every type of cancer, and different signaling pathways can regulate their maintenance and proliferation. Androgen receptor signaling plays a relevant role in the pathophysiology of prostate cancer, promoting cell growth and differentiation processes. However, in the case of prostate cancer stem cells, the androgen receptor negatively regulates their maintenance and self-renewal. On the other hand, there is evidence that androgen receptor activity positively regulates the generation of cancer stem cells in other types of neoplasia, such as breast cancer or glioblastoma. Thus, the androgen receptor role in cancer stem cells depends on the cellular context. We aimed to analyze androgen receptor signaling in the maintenance and self-renewal of different types of cancer stem cells and its action on the expression of transcription factors and surface markers associated with stemness.

## 1. Introduction

Cell replacements exist in any body system as part of homeostasis, a process driven by the generation and proliferation of new cells. Typically, cell generation is regulated by a small pool of undifferentiated cells with the ability to divide asymmetrically to generate specialized cells while maintaining the original progeny. These cells are called stem cells and have two fundamental characteristics: the ability to self-renew and high differentiation potential [1].

Stem cells can be classified based on several criteria, including potency and origin. There are four distinct types of stem cells based on their potency: totipotent, pluripotent, multipotent, and unipotent [2]. In addition, they can be classified based on their origin: embryonic stem cells (ESCs), which have pluripotency characteristics and are capable of generating cell lineages from the three germ layers; fetal and adult stem cells, which usually display multipotency capabilities while their progeny is linked to specific tissues [3]. Interestingly, cancerous cell subpopulations with characteristics similar to stem cells have been found in all cancers. These cells are called cancer stem cells (CSCs) and are characterized by being undifferentiated cells with self-renewal capacity and highly tumorigenic potential. They can also generate more differentiated cancer cells and express a phenotype like cells present in the tissue [4].

CSCs were first described in the 1990s in patients with acute myeloid leukemia using the CD34^+^/CD38^−^ cell markers, which they share with hematopoietic stem cells. In xenograft models in immunodeficient mice, CSCs repopulated all blood lineages [5]. Subsequently, CSCs were discovered in solid tumors such as breast [6], brain [7], prostate [8], and colon tumors [9]. It is currently postulated that tumors are generated by CSCs, which maintain themselves by self-renewal and are highly resistant to conventional therapies. In addition, CSCs generate progeny in a hierarchical manner due to their ability to generate more differentiated cells, which explains the high tumor heterogeneity [10]. Several signaling pathways, such as Wnt/β-catenin, TGF-β/Smad, Notch, and HedgeHog, have the ability to modulate CSCs self-renewal and proliferation [11]. Similarly, signaling mediated by sex hormone receptors has been associated with the maintenance of different populations of CSCs [12].

Androgens belong to the group of sex hormones and participate in the appearance and maintenance of male sexual characteristics. These hormones play a key role in the generation and differentiation of the prostate, as well as in the neoplasms of this organ [13]. Testosterone is the most representative androgen, exerting its primary function through the androgen receptor (AR), which functions as a ligand-activated transcription factor. Due to the action of the enzyme 5α reductase (5-αR), testosterone is metabolized into dihydrotestosterone (DHT), the most potent endogenous androgen [14]. To perform their physiological role, androgens cross the cell membrane by simple diffusion to interact with the AR in the cytoplasm, causing conformational changes that allow it to dissociate from chaperone proteins and translocate to the nucleus. Within the nucleus, the AR forms homodimers and acts as a transcription factor, binding to specific DNA sequences in the promoter regions called androgen response elements (AREs) (Figure 1a) [15]. The AR activation also regulates cellular processes through non-genomic signaling due to direct interaction in the cytoplasm with the PI3K/AKT and MAPKs pathways. Interestingly, AR expression has been found in CSCs from different types of tumors, such as breast, ovarian, and glioblastoma, demonstrating its potential to positively regulate the expression of genes associated with stemness, as well as the proliferation and self-renewal [16,17,18]. In prostate cancer (PC), the AR is preferentially expressed in cells with a higher degree of differentiation, negatively regulating the maintenance of CSCs [19]. Current evidence suggests that the AR can positively and negatively regulate processes that maintain CSCs, depending on the tumor context (Figure 1b). This review focuses on the role of androgen signaling in the self-renewal and maintenance of CSCs.

## 2. AR Signaling in Prostate Cancer

PC is the second most frequently diagnosed cancer in men, with a highly diverse incidence worldwide [20]. Similar to the healthy prostate, androgens play a crucial role in the development and progression of PC. Depending on the stage and grade of the cancer, treatment options may include active surveillance, radical prostatectomy, radiation, chemotherapy, and hormone therapy. In the case of advanced PC, androgen deprivation therapy (ADT) is usually recommended in the therapeutic scheme. To decrease androgen signaling, drugs that inhibit testosterone synthesis or act as AR antagonists are often used [21]. Despite the benefits of ADT, PC eventually develops resistance to treatment, termed castration-resistant prostate cancer (CRPC) [21,22]. When T or DHT activates the AR in PC cells, it exerts its role as a transcription factor, facilitating the expression of genes that promote cell proliferation and regulating signaling pathways that inhibit apoptosis. In CRPC, several hypotheses suggest how the AR promotes cell growth and tumor progression, even without the presence of testosterone or DHT [23]. These mechanisms include changes in the AR expression, such as its amplification and overexpression in tumor cells, the expression of splicing variants that are constitutively active, and point mutations that can modulate its activation by other ligands [24]. The androgen receptor splice variant 7 (AR-V7) is one of the most studied variants in PC, which, by having the LBD truncated, exerts its role as a transcription factor independently of activation by testosterone or DHT [25]. AR-V7 activity increases the expression of transcription factors that regulate cell cycle entry, cell growth, and proliferation. In fact, it has been pointed out as an oncological modification that promotes therapeutic resistance [26].

### 2.1. The Role of the AR in PC Tumor Growth

Cell proliferation can be regulated by the direct interaction of the AR with components of the AKT and MAPK signaling pathways, triggering cellular responses in shorter periods of time compared to the genomic pathway in PC [27,28]. Activation of the AR by its ligand positively regulates the AKT pathway through direct interaction with the p85α subunit of the PI3K protein, causing AKT phosphorylation at residues S473 and T308. Androgen-mediated activation of AKT promotes cell survival through phosphorylation of the proapoptotic proteins Bad and FKHR [27]. Similarly, AKT can positively modulate AR activation to promote cell growth. Just as activation of the AR by its ligands promotes AKT phosphorylation, activation of AKT by the Her2 signaling pathway promotes AR phosphorylation at residues S213 and S79. Thus, AKT positively modulates the AR in an androgen-independent manner [29]. Various growth factors, such as EGF and IL-8, also promote AR phosphorylation and activation under androgen-free conditions [30]. These molecules modulate the activation of the MAPK signaling cascade, causing Src kinase to phosphorylate tyrosine residues in the AR, promoting its transcriptional activity and the expression of genes that promote cell growth [30].

AR activity can modulate other signaling pathways that promote cell survival. Evidence suggests that AR activation may play a protective role in conjunction with the transcription factor FOXO3a promoting FLICE-inhibitory protein (FLIP, an inhibitor of death receptor-mediated apoptosis) expression in LNCaP cells [31] as well as a protective role for tumor cells. Similarly, TGF-β/SMAD is another signaling pathway that regulates cell growth and apoptosis in PC [32]. Androgen signaling has an essential role in regulating the activation of the TGF-β pathway because the AR suppresses the transcriptional activity of SMAD3, the main effector of the TGF-β pathway, thus negatively regulating its antitumor activity and promoting cell survival [33,34].

AR activity also promotes tumor malignancy by regulating invasion and cell migration. In LNCaP and LAPC-4 cell lines, treatments with the synthetic androgen R-1881 increased the expression of the metalloprotease MMP-2 in an AR-dependent manner [35]. Another essential protein for cell invasion is ezrin. Treatments with R-1881 also positively regulated its expression in LNCaP cells. Similarly, the AR mediates the phosphorylation of residues Thr-567 and Tyr-353, consequently promoting cell invasion in Matrigel assays [36]. Congruently, androgen signaling participates in PC epithelial-mesenchymal transition (EMT), enhancing cell migration and metastasis. Both the silencing and blocking of the AR with antagonists decreased the ability of cells to migrate and invade, a process mediated by inhibition in the expression of Slug, Snail, MMP-2, B- catenin, and vimentin, essential proteins that induce EMT in the C4-2B cell line [37]. All these data suggest that, in addition to effects on cell proliferation and death, AR activity promotes tumor malignancy by increasing migratory and invasive capacities in PC.

### 2.2. AR Expression in Prostate Cancer Stem Cells

PC stem cells (PCSCs) were isolated in 2005 by Collins et al. showing a CD44^+^/α2β1^hi^/CD133^+^ phenotype, representing only 0.1% of the cells in the tumor [38]. They also have a high capacity for proliferation and self-renewal. Currently, for the study of PCSCs, classic markers such as CD133, CD44, or α6 integrin are used in the laboratory [8,39]. Due to their properties, PCSCs are considered to play a fundamental role in the development of CRPC. The Wnt signaling pathway is important in the regulation of PCSCs as well as in the progression towards CRPC. Wnt pathway activation is associated with the increased expression of pluripotency genes such as NANOG, SOX2, and OCT4, as well as increased self-renewal capacity [40]. Even though androgen signaling plays a fundamental role in the progression of PC by promoting cell growth, migration, and invasion, most PCSCs have an AR^−^ phenotype [38,41,42]. Interestingly, studies carried out in LNCaP cell lines, LAP4 and LAP9, demonstrated that androgen depletion in the medium increased the population of cells with stem characteristics, showing a CD44^+^/CD24^−^ phenotype, as well as a high tumorigenic capacity [43].

PCSCs can reconstitute the original tumor, generating basal, luminal, and neuroendocrine epithelial cells [41]. These stem cells have an AR^−^ phenotype, but once the differentiation process is triggered, the receptor begins to be expressed, suggesting that androgen signaling favors differentiation processes over self-renewal [41]. Functional assessment in suspension cultures demonstrated that AR silencing promoted PCSCs self-renewal, increasing the number of spheres with a larger size. In contrast, AR overexpression decreased PCSCs self-renewal, showing a lower capacity for generating spheres [44]. Interestingly, bicalutamide decreased and increased the proliferation of LNCaP-CD133^−^ and LNCaP-CD133^+^ cells, respectively, suggesting that AR activation has differential effects in stem and non-stem cells [44].

AR activation has also been shown to downregulate the expression of proteins that promote stemness characteristics in PC cells. The STAT3 protein maintains pluripotency in different stem cells, promoting self-renewal processes. Schroeder et al. demonstrated that AR loss increases the expression of STAT3, allowing the generation of stemness characteristics [45]. Additionally, pharmacological inhibition of the AR promoted the expression of stemness factors such as Sox2 and CD44 [45].

AR activation can negatively modulate effector proteins of signaling pathways involved in self-renewal and proliferation, such as the Akt and Wnt pathways. Indeed, the LNCaP and C4-2 cell lines negatively regulated the expression of Akt, Wnt-1 and c-myc as well as the antiapoptotic protein Bcl-2 [44]. These data suggest that the AR downregulates stemness characteristics in both cell models. As mentioned above, patients with CRPC will eventually develop resistance to treatment. A clinical study directed by Alumkal et al. determined that low AR activity correlates with the acquisition of resistance to enzalutamide. Additionally, this low AR activity was related to the acquisition of stemness features [46]. Consistently, clinical evidence also suggests that certain CRPC-derived tumors develop lineage plasticity associated with low activity of AR signaling [47]. Vummidi et al. reported that the AR^−^ phenotype of PCSCs is mediated by the MDM2 ubiquitin ligase, which continuously degrades the AR and supports the maintenance of stemness. The loss of MDM2 induced the AR re-expression and the differentiation towards luminal cells [48]. Overall, although the AR promotes the proliferation and maintenance of PC tumors, the opposite occurs in the case of PCSCs. Our current knowledge suggests that AR expression and activity are associated with the loss of stemness characteristics; conversely, the absence of AR signaling increases the presence of PCSCs, the expression of markers associated with stemness, their clonogenicity, and self-renewal.

## 3. AR Signaling in Breast Cancer

Breast cancer (BC) is the most frequent cancer in women and the second cause of cancer death among women worldwide [49]. BC is classified according to the expression profile of the estrogen receptor α (ERα), the progesterone receptor (PR), and the human epidermal growth factor receptor (HER2). Hence, tumors that express the estrogen receptor (ER) are denominated as luminal A and luminal B. Luminal A tumors express ERα and PR but not HER2, whereas luminal B tumors express ERα, PR, and HER2, and show a higher proliferation rate than luminal A tumors; HER2-enriched tumors lack ERα and PR expression but have high HER2 expression; and triple-negative breast cancer (TNBC) tumors do not express any of the previously mentioned receptors [49]. Interestingly, the AR is expressed in 60–80% of mammary gland tumors, mainly among Luminal A neoplasias [50].

The effects of the AR on cell proliferation and tumor progression are unclear because its action depends on the co-expression of ERα in BC. In ERα positive tumors, AR expression has been associated with a longer life expectancy and having smaller and lower-grade tumors [51]. However, a high AR:ER ratio is associated with the development of resistance to treatment, which suggests that the expression level of both receptors could influence tumor growth [52]. In the case of tumors that do not express ERα, the AR exerts mainly a protumoral activity [53,54].

### 3.1. Action of Androgens in ERα^−^ and ERα^+^ Tumors

In neoplasias lacking ERα, androgens stimulate cell proliferation and tumor growth. Experimentally, the AR can regulate the Wnt signaling pathway, which plays an essential role in cell proliferation [54,55]. Moreover, the AR interacts with β-catenin to modulate gene expression, enhancing the transcription of the CMYC oncogene, and promoting cell growth processes [54]. Similarly, in the MDA-MB-453 cell line, AR activation through DHT promotes WNT7B transcription, an essential mediator in the Wnt activation. In this model, transcriptional complexes comprising AR, β-catenin, and FOXA1 promoted HER3 expression [55]. HER3 can form heterodimers in HER2^+^ BC cells, driving cell proliferation via the PI3K/AKT pathway [56]. Bicalutamide treatment reduces HER3 expression and AKT phosphorylation [55]. Similarly, enzalutamide treatment and AR silencing in SKBR3 and HCC1954 cell lines decreased HER2 phosphorylation without affecting the HER2 and HER3 protein content. This suggests that the AR exerts non-genomic action in HER2^+^ overexpressing cells, regulating downstream pathways, such as PI3K and ERK [51]. Notably, inhibiting the PI3K pathway leads to decreased AR expression [57].

In TNBC cells, activation of the G-protein-coupled estrogen receptor (GPER) exerts antitumor activity [58]. In MDA-MB-231 and Hs578T cell lines, DHT promotes cell growth and viability while suppressing GPER expression at both transcript and protein levels. Notably, GPER activation inhibited AR-mediated cell growth [59]. In addition to promoting cell proliferation in ERα^−^ cancer cells, there are reports of its participation in cell migration. Once the AR is activated, it can produce complexes with Src kinase and PI3K, which generate structural changes that promote cell motility and invasion [60]. Overall, in ERα^−^ BC cells, AR activity promotes malignant processes by regulating essential signaling pathways in cell proliferation, as well as in cell migration and invasion processes.

In BC tumors that express ERα, estrogens play an important role in cell proliferation and tumor growth. Certainly, several of these tumors co-express other sex hormone receptors, such as the PR and AR. ERα promotes gene expression in signaling pathways that promote cell growth in cancer cells [51]. The AR activation is associated with antiproliferative activity in these tumors. Androgen administration decreased cell growth in in vitro models. DHT treatments in the MCF-7 cell line decreased cell proliferation due to reduced cyclin D1 expression provoked by repression in the CCND1 gene transcription mediated by the AR [61]. Similar results were observed in aromatase expression, which is repressed by the AR [62], causing less estrogen synthesis from testosterone. In this cellular model, the repressor action on AR transcription was mediated by DAX1 [61,62].

AR expression has been associated with a longer life expectancy and with having smaller and lower-grade tumors. The AR exerts antiestrogenic activity due to the competition with the ERα for their hormone response elements in the promoter region of their target genes. Thus, the AR inhibits the gene expression that promotes cell proliferation regulated by the ERα genes [63]. In addition, the AR can bind to the AREs sequences in the ERβ promoter, inducing its expression and provoking an antitumoral activity [64]. These data suggest that the antitumoral activity of the AR in ER+ tumors could be regulating ERα and ERβ activity.

### 3.2. Regulation of Breast Cancer Stem Cells by the AR

BC stem cells (BCSCs) were the first cancer stem cells reported in solid tumors. For their isolation, Al-Hajj et al. identified a cell population in mammary tumors that expressed the CD44^+^/CD24^−^ phenotype, with a very high tumorigenic capacity, inducing tumors with 200 cells inoculated in immunosuppressed mice [6]. Sex hormone receptors, such as the ER, PR, and AR, are also expressed in this population. In TNBC cells, enzalutamide decreased the ability to generate colonies on soft agar, suggesting a lower growth under anchorage-free conditions. In addition, the drug decreased cell viability and increased apoptosis and necrosis in in vivo models [65]. A higher expression of the AR was also found in a study with cells in forced suspension to increase the population of BCSCs [64]. AR silencing decreased both the ALDH^+^ cell population from 55 to 40% in cells of the SUM159PT line as well as the generation of three-dimensional structures that allow the enrichment of BCSCs, called mammospheres. These results were also observed with enzalutamide. In the MDA453 cell line, AR silencing increased the percentage of CD24^+^ cells, indicative of a differentiated phenotype, and reduced the efficiency of mammospheres derivation [66]. It is interesting to note that, in a study carried out on 197 patients, the loss of the AR correlated with the presence of a stem phenotype (CD44^+^/CD24^−^) in TNBC [67]. Fernandez et al. determined that the AR is a positive modulator of the transcription factor RUNX1 at transcript and protein levels and its transcriptional activity is related to the expression of genes associated with stemness such as KLF4, OCT4, and SOX4 in TNBC [68]. A study by Rosas et al. demonstrated that activation of the TGF-β signaling pathway increased AR expression under anchorage-independent conditions. Consistently, AR interacted with TGFB1 and SMAD3 regulatory sequences. Inhibition of both TGF-β and AR decreased survival in anchorage-independent conditions [69]. Additionally, the transcription factor SNAI1 is important in the epithelial–mesenchymal transition. In a study carried out by Tsirigoti et al. it was determined that the loss of SNAI1 increased cellular plasticity in TNBC cells. This effect was due to greater expression and activity of FOXA1 and AR, promoting a higher differentiation potential [70]. In summary, the experimental evidence suggests that the activity of the AR in cancer stem cells present in TNBC tumors positively regulates the self-renewal and the expression of markers associated with stemness; however, it is necessary to evaluate the actions of the AR in ER-positive tumors to establish if it has a role in the maintenance of BCSCs, evaluating the expression of stem genes, self-renewal and tumorigenesis.

## 4. AR Signaling in Glioblastoma

Gliomas are tumors of the central nervous system (CNS) of glial origin. Glioblastoma (GB) is the most frequent and lethal, with an incidence of 3.2 per 100,000 inhabitants, more frequent in males than in females, with a ratio of 3:2 [71]. This tumor rarely metastasizes outside the CNS; however, it is highly infiltrative in the brain parenchyma, generating diffuse tumors with indefinite borders. The higher incidence in men than women suggests that sex hormones play a role in the prevalence of this cancer. There is evidence that patients with GB have a higher serum testosterone concentration than healthy subjects, regardless of gender [72]. Consistent with these data, tumor tissue samples from GB patients showed increased AR expression compared to peripheral normal brain tissue. Indeed, AR expression has also been found in the human GB cell lines A172, LN18, LN229, M059, T98G, U87MG, U118MG, and U138MG [73].

Several studies suggest that AR activation in GB cells can occur in both androgen-dependent and independent mechanisms. In vitro studies showed that testosterone or DHT treatments promote cell proliferation, migration, and invasion of human GB-derived cell lines through AR activation [74,75]. Concordantly, the activation of the AR by DHT in the U87-MG cell line hindered the antiproliferative effects produced by the TGF-β signaling pathway. These effects resulted from the AR binding to SMAD3, the primary effector of this pathway, which prevented SMAD3 translocation into the nucleus and its subsequent activity [73]. In the U87 cell line, androgen-independent AR activation was observed, wherein EGFR signaling induced AR phosphorylation and its translocation to the nucleus to execute its functions [76]. Additionally, the presence of the splicing variant AR-v7 was reported. This variant lacks the ligand-binding domain, leading to the constitutive activation of the AR in an androgen-independent manner, thereby promoting tumor growth [77]. All these data suggest that AR activation, either in an androgen-dependent or independent manner, promotes GB cell proliferation and plays a critical role in its progression.

### AR Activity in the Maintenance of GSCs

Glioma stem cells (GSCs) have been identified in GB, displaying high proliferation and the ability to self-renew, generate new tumors, and differentiate into cells of neuroglial lineages. The pioneering work of Sheila Singh and her colleagues involved isolating GSCs through the membrane marker CD133 [7]. Subsequently, different markers have been utilized for their study and isolation, including CD15 [78], α6-integrin [79], Sox2 [80], Oct4 [81], Nanog [82], and ALDH1A3 [83], among others. There is evidence that mutations in neural stem cells (NSCs) can give rise to GSCs, making NSCs a potential origin for GSC [84]. In physiological contexts, NSCs participate in developing the CNS during embryogenesis and neurogenesis in adult mammals [85]. Both adult and embryonic NSCs express an AR. Recent evidence suggests that testosterone and DHT promote the proliferation and self-renewal of NSCs generated from the H1 and H9 human embryonic stem cell lines. Private androgens facilitated the differentiation of NSCs into excitatory neurons in an organoid model [86]. DHT treatments also enhanced the self-renewal of neural progenitor cells derived from mouse embryos, leading to an increase in the number of neurospheres in the culture and the expression of the stemness marker ALDH1A3. Interestingly, a more pronounced limitation to neuronal differentiation was observed in the presence of DHT [87]. These findings suggest that androgens promote NSCs proliferation and self-renewal while limiting their differentiation capabilities. Similar trends were noticed in GSCs. The utilization of bicalutamide and enzalutamide reduced the ability of human GB-derived cells to generate neurospheres and the proportion of CD133^+^ cells within the culture [18]. Congruently, treatment with the 5α-reductase inhibitor, finasteride, decreased sphere formation in the U373 and T98G cell lines [88]. The decline in the number and size of neurospheres implies that AR activity regulates GSC self-renewal. Additionally, the inhibition of AR activity reduced the expression of stemness markers such as NANOG, OCT4, and SOX2 [18,88]. In BC models, androgen signaling has been linked to the Wnt pathway, demonstrating the AR interaction with β-catenin to regulate gene expression. Finasteride treatments in GSCs lowered β-catenin expression, suggesting that androgens could regulate Wnt signaling similar to that reported in BC, thus contributing to stem cell maintenance [88]. Currently, experimental evidence suggests that AR activation is a crucial player in maintaining stem cells within GB, fostering the expression of stemness factors and self-renewal. However, more studies are required to validate these observations. In common with BCSCs, the AR in GB seems to support stemness contrasting with the observed role in PCSCs.

## 5. AR Signaling in Other CSCs

As has been observed in BC and GB, there is evidence that AR signaling can positively regulate stemness characteristics in other CSCs. In the case of cells derived from hepatocarcinomas, androgens regulate stemness [89,90]. Agriesti et al. demonstrated that treatments with the anabolic steroid nandrolone in the HepG2 cell line increased the stem-like phenotype in cell models, increasing the percentage of CD133^+^ cells. Similarly, other markers associated with stemness, such as Myc, Lin28, Nanog, Klf4, and Sox2, also increased their expression [89]. Additionally, these authors reported similar results in non-cancerous stem cells, both hematopoietic and mesenchymal, in which nandrolone increased stemness and decreased the differentiation capacity. Consistently, Jiang et al. demonstrated that DHT treatments increased clonogenicity and the sphere generation capacity as well as the expression of the transcription factors Nanog, Oct4, and Sox2 in various hepatocarcinoma-derived cell lines. It was also determined that the regulation of stemness by AR signaling was dependent on the Nanog action [90].

Similar results were observed in cells derived from ovarian cancer. The relationship between AR signaling and Nanog was investigated by Ling et al. demonstrating that DHT treatments increased the transcriptional activity of the gene encoding Nanog and the clonogenic capacity in ovarian CSCs. The expression of other pluripotency genes such as Sox2 and Oct4 was also increased by DHT [16]. In analyses reported in cellular models derived from human teratocarcinomas, AR overexpression increased CD133^+^ cells and spheres generation. Furthermore, AR silencing decreased stemness characteristics [91]. Interestingly, AR signaling, but not DHT treatments, promoted cell growth, suggesting that AR activity is ligand-independent. Consistent with these reports, in the case of endometrial cancer, AR expression increased the percentage of CD133^+^ cells and the expression of other surface markers and transcription factors associated with stemness [92]. Finally, Chen et al. showed solid evidence demonstrating that the AR increased the population of CSCs, their self-renewal, and clonogenicity in bladder cancer, an effect inhibited by enzalutamide. The AR was also related to the positive expression of the transcription factors Sox2, Bmi1, and Nanog [93]. Current evidence, although limited, indicates that AR activation in CSCs from different cancers can regulate the expression of factors associated with stemness and their self-renewal capacity (Table 1).

## 6. Conclusions and Perspectives

Cancer stem cells are a highly researched topic in oncology due to their remarkable ability to sustain tumor growth through self-renewal and differentiation. These cells resist conventional therapies and are closely linked to tumor relapse. While the discovery of cancer stem cells is not recent, an effective strategy to inhibit their growth in several cancer types remains elusive. Although the AR plays a key role in the development and progression of PC, its role in regulating the maintenance of PCSCs appears to be the opposite. It is interesting to note how PCSCs usually have an AR^−^ phenotype, showing a high capacity for self-renewal. On the contrary, when AR expression is induced, PCSCs decrease their capacity to generate spheres in culture and the expression of stemness markers such as CD44 and Sox2. Congruently, when PCSCs begin to differentiate, AR expression begins to increase, suggesting that in the case of PC, AR signaling favors differentiation over self-renewal of CSCs.

In contrast, AR activity is related to the maintenance of stemness in glioblastoma, hepatocellular carcinoma, breast, ovarian, endometrial, and bladder cancer. Interestingly, in CSCs of these neoplasia, AR activity increased the expression of the genes associated with pluripotency such as Sox2, Oct4, and Nanog. Nanog showed an important relationship with AR signaling for the maintenance of stemness in hepatocellular and ovarian cancer. Moreover, the expression of surface markers closely related to stemness phenotypes also increased due to AR activity. In the case of glioblastoma and breast cancer, the use of AR antagonists decreased the expression of factors associated with stemness.

Consistently, AR activity increased the clonogenic and self-renewal capacities of CSCs from glioblastoma, hepatocellular carcinoma, breast, ovarian, endometrial, and bladder cancer, contrasting with what was reported in PC. In cell suspension models, AR signaling generated a greater number and diameter of spheres. Similarly, the percentage of CD133^+^ cells, indicative of stemness, also increased. Importantly, the AR can cross-link with signaling key pathways for the maintenance of stemness, such as those regulated by TGF-β and Wnt/β-catenin, and pathways involved in cell proliferation and growth mediated by AKT and MAPK. Although more experimental evidence is still needed, current findings suggest that the AR plays a key role in the regulation of various CSCs, and should be a target for the treatment of various cancers.

## Figures and Tables

**Figure 1 ijms-24-15085-f001:**
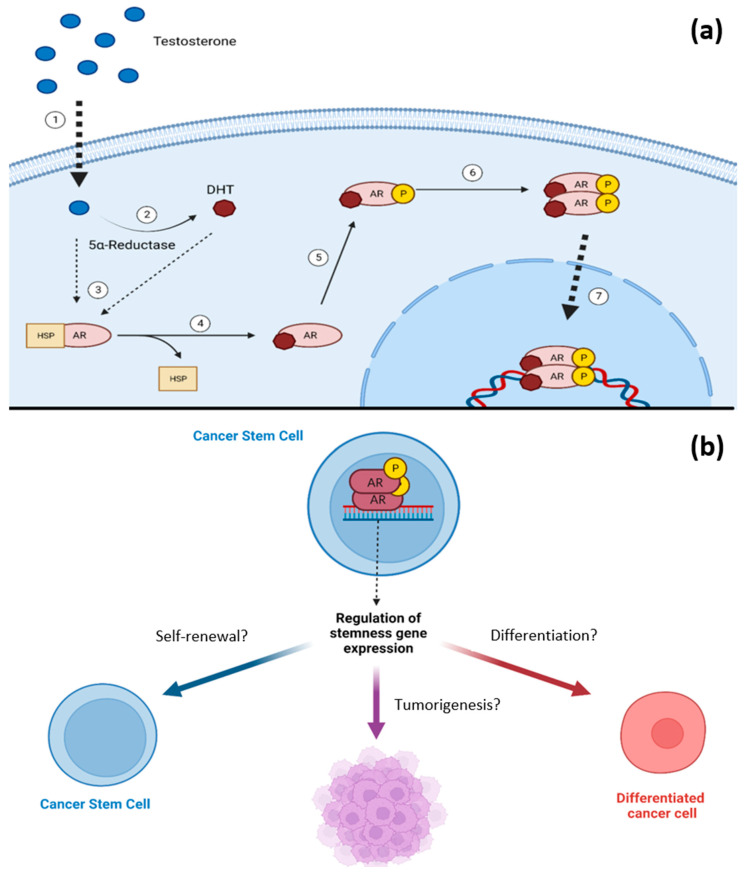
Androgen receptor signaling in cancer stem cells. (**a**) Testosterone, due to its hydrophobic characteristics, crosses the cell membrane ①. Once in the cytoplasm, it can be metabolized into DHT ② or directly bind to the AR ③. The interaction of the AR with its ligand generates conformational changes that release it from heat shock proteins ④ that keep it stable in the cytoplasm and protect it from degradation. Once the AR is bound to its ligand, it can be phosphorylated ⑤ and subsequently form a homodimer ⑥, which can translocate to the nucleus and recognize specific gene sequences ⑦. (**b**) The activated AR in CSCs may promote the expression of genes associated with stemness. However, its effects on self-renewal, differentiation, and tumorigenic capacity seem to depend on the tumor context. Created with Biorender.com.

**Table 1 ijms-24-15085-t001:** Effects of AR signaling on stemness maintenance.

Cancer Stem Cell	AR Effect on Stemness	References
Prostate Cancer Stem Cell	Decreases self-renewal capacity Decreases expression of stemness factors Increases cell differentiation	[42,44,45]
Breast Cancer Stem Cell	Promotes self-renewal Promotes expression of stemness factors	[65,66,68]
Glioma Stem Cell	Promotes self-renewal Promotes expression of stemness factors	[18,88]
Hepatic Cancer Stem Cell	Promotes self-renewal Promotes expression of stemness factors	[89,90]
Ovarian Cancer Stem Cell	Promotes self-renewal Promotes expression of stemness factors	[16,91]
Endometrial Cancer Stem Cell	Promotes self-renewal Promotes expression of stemness factors	[92]
Bladder Cancer Stem Cell	Promotes self-renewal Promotes expression of stemness factors	[93]

## Data Availability

Not applicable.

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
