# Peer review of "Cancer Stem Cells and Androgen Receptor Signaling: Partners in Disease Progression"

_ijms, 2023, doi:10.3390/ijms242015085_

Round 1
Reviewer 1 Report
The review by Quintero et al. summarizes in a very good the knowledge about androgen receptor-mediated control of stemness and stem cells in cancer. Authors provide the knowledge of androgen regulation of CSC in three different cancers.
The review is well written and covers an important topic of hormonal control of stem cells..
However, some additional and important contributions within this should be included. This covers some papers and also some knowledge in other cancers in which androgens control the expression of important stem cell markers.
Please include following contributions:
1. Prostate cancer:
doi: 10.1073/pnas.1922207117.
doi: 10.1038/s41467-022-32701-6
2. Brast Cancer
doi: 10.1038/s41419-022-05280-z.
doi: 10.3389/fonc.2020.01083.
3. Glioblastoma
doi: 10.1210/endocr/bqac002.
Please add knowledge about stemness control by androgens in these two other cancer types.
4. Hepatocellular cancer:
doi: 10.18632/oncotarget.9192.
doi: 10.1038/s41598-020-58871-1.
5. and Endometrial cancer
Although not much is known here, some stemness genes may be regulated by androgens
well written
Reviewer 2 Report
The review manuscript by Quintero is a very interesting and laudable attempt to delineate the relationship between AR and cancer stemness. Though at its current version it lacks solid evidence to It brings to the point of there is a correlation between AR expression and stemness – however as the relation in different cancer is varying, it is not clear with the current level of evidence it is purely correlative or some causal relationship. The authors need to provide substantial literature evidence to show the relationship between AR and stem cells are different and context dependent.
Abstract should be rewritten to capture the content of the review.
Perspective section should capture the authors view based on the literature evidence.
Minor – typos. For ex. Despite our current knowledge, extensive studies on various types of cancer are essential to elucidate the regulatory role of RA in the maintenance of stemness.- is it RA or AR?
No specific comments
Round 2
Reviewer 1 Report
Authors addressed criticism in full satisfactory manner.
Reviewer 2 Report
The revised version improved substantially. I have no more comments